# A multi-step analysis and co-produced principles to support equitable partnership with Liverpool School of Tropical Medicine, 125 years on

**Robinson Karuga[1‡], Rosie Steege[2‡], Shahreen Chowdhury[2]\*, Bertie Squire[2], Sally Theobald[2‡], Lilian Otiso[1‡]**

1 Department of Research and Strategic Information, LVCT Health, Nairobi, Kenya, 2 Department of International Public Health, Liverpool School of Tropical Medicine, Liverpool, United Kingdom

‡ RK and RS are joint first authors on this work. ST and LO are joint last authors on this work.
\* shahreen.chowdhury@lstmed.ac.uk

## Abstract

Transboundary health partnerships are shaped by global inequities. Perspectives from the "global South" are critical to understand and redress power asymmetries in research partnerships yet are not often included in current guidelines. We undertook research with partners working with the Liverpool School of Tropical Medicine (LSTM) to inform LSTM's equitable partnership strategy and co-develop principles for equitable partnerships as an entry point towards broader transformative action on research partnerships. We applied mixed-methods and participatory approaches. An online survey (n = 21) was conducted with LSTM's transboundary partners on fairness of opportunity, fair process, and fair sharing of benefits, triangulated with key informant interviews (n = 12). Qualitative narratives were analysed using the thematic framework approach. Findings were presented in a participatory workshop (n = 11) with partners to co-develop principles, which were refined and checked with stakeholders. Early inclusion emerged as fundamental to equitable partnerships, reflected in principle one: all partners to input into research design, agenda-setting and outputs to reflect priorities. Transparency is highlighted in principle two to guide all stages including agenda-setting, budgeting, data ownership and authorship. Principle three underscores the importance of contextually embedded knowledge for relevant and impactful research. Multi-directional capacity strengthening across all cadres is highlighted in principle four. Principle five includes LSTM leveraging their position for strategic and deliberate promotion of transboundary partners in international forums. A multi-centric model of partnership with no centralised power is promoted in principle six. Finally, principle seven emphasises commitment to the principles and Global Code of Conduct values: Fairness, Respect, Care, Honesty. The co-developed principles are part of ongoing reflections and dialogue to improve and undo harmful power structures that perpetuate coloniality within global health. While this process was conducted with LSTM-Liverpool's existing partners, the principles to strengthen equity are applicable to other institutions engaged in transboundary research partnerships and relevant for funders.

**Data Availability Statement:** Excerpts of our data are available within the manuscript. Due to the sensitive nature of the research, further excerpts

other than provided in the manuscript, would have to be significantly redacted to remain anonymous as the data contains potentially identifying information. This would violate the data sharing agreement to which participants consented and as agreed by the Liverpool School of Tropical Medicine Research Ethics Committee. Requests for redacted data may be sent to lstmrec@lstmed.ac. uk,

**Funding:** This study was supported by Liverpool School of Tropical Medicine (LSTM) core funds in the form of salaries to the authors who are faculty at LSTM. Non-LSTM faculty members RK, LO, and RS (when she was not employed at LSTM) received some financial support for undertaking the research. The specific roles of these authors are articulated in the "author contributions" section.

**Competing interests:** The authors have declared that no competing interests exist.

# Introduction

## Inequities in global health research partnerships

Transboundary partnerships in global health are inherently complex and shaped by global inequities. These inequities are often reflected within research partnerships due to the imbalance of economic and academic resources, which disproportionately advantage those working in higher income settings [1–4]. Inequities in partnerships are rooted in socio-political and historical structures of inequity and colonialism but are maintained by the current 'status quo' in global health research structures and funding steams. Global health funding predominantly comes from high income countries (HIC)–the global North—and partners in the global North are often the lead grant recipients [3]. As such, Northern partners may hold power over research conducted in, and for, Low- and middle-income countries (LMICs)—the global South. Literature shows how the allocation of funding can contribute to unequal decision-making and division of labour [5–8]. Northern researchers often have more opportunities to shape research and provide intellectual inputs, and at worst appropriate local data. Khan *et al.* write that global health organisations may "*perpetuate the power imbalances they claim to rectify through colonial and extractive attitudes, and policies and practices that concentrate resources, expertise, data and branding within high-income country institutions.*" [9] (p.1). It has been posited that when transboundary partnerships prioritise research needs from the global South, and embed research capacity development within partnership aims, those with fewer resources can benefit from such collaboration [10]. However, Ishengoma (2016) argues that "*despite having the potential to enhance the research capacities of universities and individuals, North–South research collaborations have had limited impact because of the neocolonial nature of the donor–recipient framework*" [8] (p.150).

Recognising that language has the power to harm and reinforce systemic injustice, semantics require critical reflection and adaptability [11, 12]. In this paper, we mainly refer to transboundary research partnerships, which recognise the complexity of these partnerships and is described by the Swiss Commission for Research Partnerships with Developing Countries as partnerships that–'*cross economic, social, and cultural borders or divides–in short, they are transboundary in various dimensions*' [13]. However, we also use global North and global South when we wish to discuss the location of researchers, whilst recognising the inaccuracies of this term. We also acknowledge that "Tropical Medicine" is another homogenising and problematic term [14].

## Liverpool School of Tropical Medicine (LSTM): Looking backwards and looking forward

2023 marked 125 years of Liverpool School of Tropical Medicine (LSTM), the world's oldest school of Tropical Medicine, created after an appeal from Joseph Chamberlain, Secretary of State for the colonies to address "mortality and morbidity arising from endemic 'tropical diseases'" [15]. The intention was to advance teaching and research on tropical medicine in Britain and facilitate political and economic exploitation by Western powers and expand European colonial empires [15–19]. As argued by Burgess (2022), we need research, and researchers, to acknowledge how the past governs and shapes our efforts to change health [20]. Thus, this landmark calls for a critical and collective reflection of an organisation rooted in colonial and tropical medicine. Staff in institutions such as LSTM- Liverpool, must critically reflect on how its history shapes inequalities today, and take action to disrupt current knowledge asymmetries through sharing relative power, examining working practices, languages used and learning about what matters to partners in working relationships. LSTM—Liverpool

is currently engaged with transboundary partners through: established research and education collaborations, involvement in organisation development, having staff embedded in overseas organisations and overseas LSTM offices and is committed to fostering equitable global partnerships in research, education and organisational development. We define partners as organisations that have a pre-existing collaboration with LSTM—Liverpool and engaged in one or more of the following: research, teaching, policy or practice.

## Looking forward: Transformative research partnerships in global health

There is now no debating the need for change within the discipline of global health [20]. We are at a tipping point [21]. There are calls to level the research playing field for research and promote epistemic diversity by '*aligning the positionality and the gaze of global health funding models*', ([23], p.7) enhancing representation of researchers from Southern settings through diverse research teams and overhauling editorial structures that restrict access to knowledge and privilege high-impact journals in the global North [3, 22, 23]. Aboderin *et al.* (2023) argue for a profound rebalancing of the relationships between Africa and the Global North and worldwide science and research ecosystems [24]. Change however, involves the dismantling of deeply entrenched power structures. This is no small task; power imbalances manifest in a myriad of intersecting ways [24]. As both Burgess and Weick have written however, social change can happen by 'small wins'—tangible acts that add up to more than the sum of their parts [20, 25]. One tangible act in the field of global health is the promotion of equitable partnerships, although as cautioned by Aboderin *et al.* (2023) this needs to be viewed as an entry point to transformative collaborations and a wider balancing of power in the global scientific effort [24].

The recent explosion in guidelines for equitable partnerships speak to efforts being made at an operational level to work towards equity. Despite this, Southern authors' perspectives are often missing or under-represented in guideline development [2, 26, 27]. A recent review by Voller et al. (2022), found only two guidelines [28, 29] out of 22 identified documents were developed predominantly or exclusively drawing on Southern stakeholders as participants [2]. This perpetuates colonial structures of knowledge generation and demands attention. As Audre Lorde famously wrote '*the master's tools will never dismantle the master's house*' [30]—a powerful metaphor which has been recently applied to decolonising global health [31].

Aligned with efforts to decolonise global health, we aimed to co-create institutional guidance with transboundary partners to promote mutual learning, inform practice of equitable partnerships and co-develop a set of principles for equitable partnerships to support accountability and promote trust. The research was conducted with LSTM-Liverpool's partners in the global South to foreground their perspectives and co-create principles to strengthen equity. However, these principles are applicable beyond LSTM–to other institutions engaged in transboundary research partnerships, and development partners/funders working towards equitable partnerships.

## Methods

We used mixed methods and participatory research approaches to co-create knowledge, the principles and encourage critical reflection and dialogue. We collected data from October 2021 to November 2022 and engaged individual respondents from 20 transboundary partner organisations, across 15 different countries in Africa, Asia and the Middle East. Organisations included other academic and research institutions, non-government organisations and community-based organisations in the global South. We employed several methods in succession (Fig 1). These individual methods were designed to feed into one another and helped to

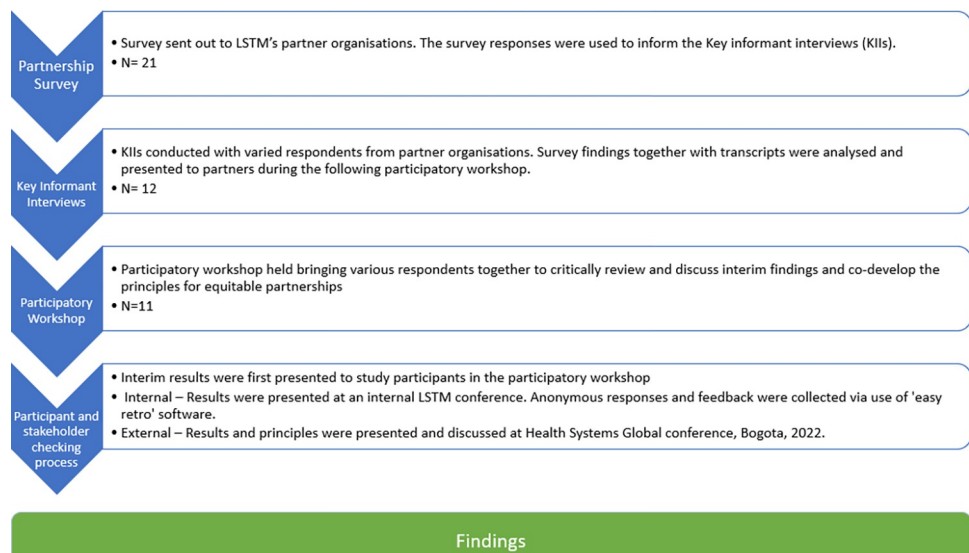

**Fig 1. Equitable partnerships study process–demonstrates how each consecutive stage of process informed the following stage.**

triangulate findings across multiple methods and perspectives, over a period of six months. These steps are described in more detail below.

## Step one

We developed an online survey questionnaire for use with transboundary partner organisations to seek anonymous responses. The aim was to elicit an understanding of what LSTM and partners do well and where LSTM and partners have gaps in equitable engagement. The survey was based on, and adapted, questions from the three domains listed in the Research Fairness Initiative's (RFI) reporting guide [32]. The guide seeks to improve the fairness, efficiency and impact of research collaborations globally, though we recognise this is a framework developed in the North, it served as a standardised starting point. The domains are 1. Fairness of opportunity (before research), 2. Fair process (during research) and 3. Fair Sharing of Benefits, Costs & Outcomes (after research) [32]. The questionnaire can be found in S1 Text LSTM Partnership Survey and includes Likert scale questions and free text.

*Recruitment.* We approached organisations which have a pre-existing partnership with LSTM (n = 39). We collaboratively developed a table of these organisations across LSTM to ensure we captured a range of partnerships–from multiple contexts, areas of focus, type of partnership (e.g teaching or research) and length of engagement (long term engagement with LSTM vs short term /newer partnerships).

The ethical process required an email introducing the study to be sent from LSTM-Liverpool's Dean of Clinical Sciences and International Public Health (BS) to senior leads at partner organisations to first gain their approval to participate in the study. If they wished to participate, we invited them to respond directly to RK or RS confirming approval. RK and RS sent a survey link that they were asked to cascade within their organisation to capture diversity. We had 21 responses, from eight organisations working across 11 countries. The requirement to approach senior leads to cascade the invitation relied on their active engagement and may have introduced some bias and contributed to a low response rate. We discuss this further in the limitations section. These responses then informed the discussion guides for the Key informant interviews (KIIs).

## Step two

We conducted qualitative interviews with purposively sampled key informants (n = 12), to gain diversity in organisations (long terms vs. short term engagement with LSTM), gender, level of seniority and a range of geographical contexts (KIIs were based in Africa and Asia, no partners from the Middle East were available to participate). We co-developed the interview guides across LVCT Health (Kenya) and LSTM (UK) and focused on several areas relevant to partnerships such as research agenda setting, impact of the partnership on global and local policies, funding structures, capacity development initiatives and language concepts. Interviews triangulated findings from the survey and explored issues in more depth to understand barriers and enablers to support equitable engagement.

*Recruitment*. Survey respondents were also asked to indicate if they or another member of their team would like to participate in a KII and we recruited some participants this way. Additionally, we sent emails to gauge interest to participate in interview from those who had not participated in the survey but had indicated willingness to participate in the study. RS and RK conducted the interviews virtually via Microsoft Teams or Zoom and obtained informed consent. While authors had access to identifiable information about participants due to the participatory nature of the study, all data was anonymised after data collection.

## Step three

We held a collaborative workshop in March 2022 with LSTM—Liverpool's partners working in the global South (n = 11) to co-develop a set of principles for equitable partnership (please see Box 1 for final principles). We sent invitations to an online co-development workshop to

### Box 1. The co-developed principles for equitable partnership

**Co-Developed Principles for Equitable Partnerships**

1. Opportunity for all partners to input into research design, agenda setting and lead in development of outputs to ensure it reflects the priorities and needs agreed by all partners.

2. Transparency to guide all stages of the partnership from agenda setting, budgeting, data ownership, authorship, training and education.

3. Recognition that relevance is key to shaping agendas and conducting research that is appropriate and impactful in research settings.

4. Acknowledgement that professional development at all levels/cadres requires mutual, multi-directional capacity strengthening and exchange based on development needs assessed and identified by all partners at the outset.

5. Commitment to deliberate and strategic promotion of leadership of LMIC partners (where appropriate) in collaborations with LSTM (e.g., grants, capacity strengthening).

6. Commitment and adherence to a multi-centric model of partnership–not necessarily LSTM-Liverpool at centre, no centralized power, shared responsibilities.

7. Commitment to the four values of the Global Code of Conduct within all collaborations: Fairness, Respect, Care, Honesty and pay attention to institutional values.

all organisations who agreed to partake in the study, this meant we had some attendance from people who were interviewed or participated in the survey but largely attendance was from people who we had not previously spoken to, increasing diversity of respondents. The workshop allowed us to disseminateour interim study findings and begin the process of defining a set of principles equitable engagement with transboundary partners. This was facilitated by external partners at LVCT Health, Kenya (LO, RK) and staff who were not within senior LSTM management (RS, SC). Senior LSTM colleagues (SBS, ST) joined at end of the session to affirm their commitment to the process and give thanks for partners' participation in the process.

*Recruitment*. We advertised a date, time and meeting invitation via email to all organisations who had indicated willingness to participate in the study.

## Step four–participant and stakeholder checking

Following participant checking with partners during the participatory workshop, we shared and sought feedback on the findings and draft principles at an internal LSTM staff hybrid conference (n = 173) (that included staff based in Liverpool and Kenya, Malawi, Nepal, Uganda and Zimbabwe). We collated inputs and comments via easyretro (https://easyretro.io/), which informed our thinking around making the principles actionable and accountable. We also undertook an external feedback exercise via a 'world café' approach at Health Systems Research (HSR) Conference, Bogotá November 2022 (approximately 40 participants). This provided opportunity to present the co-developed principles to a wider audience for discussion and refinement. There was consensus that the principles were appropriate but questions were raised about accountability, including processes and resources for monitoring and learning, which we set out in the discussion. Finally, we shared the amended co-developed principles with all study participants for final comments.

## Data analysis

We used simple descriptive statistics to identify key trends in the survey data. Interviews were transcribed verbatim and collated with free text responses from the survey. We organised, coded and summarised data according to themes using an inductive framework approach. Coding was facilitated by NVivo 12 (QSR International, Australia). This allowed us to explore and explain links between codes and themes [33]. The participatory workshop with transboundary partners (n = 11) served as participant checking on the analysis as well as an opportunity for co-creation of the principles. Three researchers with varying positionalities came together to validate coding structures and summarise themes (RS, SC, RK).

## Ethical considerations

This study received approval from the Research Ethics Committee at LSTM (21–060) in September 2021. Participant information sheets were provided and written and/or verbal informed consent was obtained for all survey responses, KIIs and workshop participants. Additional information regarding the ethical, cultural, and scientific considerations specific to inclusivity in global research is included in the S1 Checklist.

## Positionality, power and engagement

The survey allowed for anonymous responses to be gathered so participants could be critical. The positionality of interviewers was also important to mitigate against bias–interviews were conducted by LVCT Health staff member RK, a Kenyan man based within an organisation

external to LSTM, or a previous LSTM staff member RS, a British woman who had not been part of senior management at LSTM and was independent from LSTM at the time of interviews; both interviewers felt that they were able to build rapport and have open and critical conversations. Further, a general positive perception respondents held of working with LSTM-Liverpool may have facilitated critical discussion of the power imbalances within global health research. We felt participants were comfortable to discuss the uncomfortable, which was essential to have meaningful dialogue on this. As a study team we represent different genders, citizenships and races and recognise that we occupy elite positions within our societies: working in a higher education institution in the UK, and an established NGO in Kenya. We acknowledge that our interpretation of the data is based on our various identities, and experiences of power. See our reflexivity statement (S3 Text Consensus statement).

## Results

We gathered responses from participants working across 20 different organisations and 15 different country contexts across the stages of the research process (Table 1).

Results are presented first by findings on language and terminology, then by thematic areas mapping to the domains from the RFI–fair opportunity, fair process and fair benefit sharing. We end with co-created principles and a discussion of how these were shaped. Under each theme, findings are integrated and triangulated across the survey, interviews and participatory workshop, including recommendations from partners where applicable.

### Language and terminology

Overall, while there was a preference for the term 'global South' there was no consensus. One participant emphasised that different UN agencies and multilaterals all have different

**Table 1. Respondent characteristics across the three stages of the research process.**

| Method | Survey (n = 21) | KIIs (n = 12) | Participatory Workshop (n = 11) |
|---|---|---|---|
| Organisations | N = 8 | N = 12 | N = 10 |
| Gender | Man (n = 13) | Man (n = 9) | Man (n = 8) |
| | Woman (n = 7) | Woman (n = 3) | Woman (n = 3) |
| | Other identity (n = 1) | Other identity (n = 0) | Other identity (n = 0) |
| Geographical Region | North Africa (n = 1) | North Africa (n = 0) | North Africa (n = 1) |
| | East Africa (n = 6) | East Africa (n = 5) | East Africa (n = 4) |
| | West Africa (n = 2) | West Africa (n = 2) | West Africa (n = 3) |
| | Southern Africa (n = 1) | Southern Africa (n = 2) | Southern Africa (n = 1) |
| | South Asia (n = 1) | South Asia (n = 3) | South Asia (n = 0) |
| | Middle East (n = 0) | Middle East (n = 0) | Middle East (n = 1) |
| | Europe (n = 1) | Europe (n = 0) | Europe (n = 0) |
| | Dual (n = 1) | Dual (n = 0) | Dual (n = 0) |
| Organisations' involvement with LSTM | 2–5 years (n = 4) | 2–5 years (n = 1) | 2–5 years (n = 1) |
| | 5–10 years (n = 11) | 5–10 years (n = 5) | 5–10 years (n = 4) |
| | 10–20 years (n = 2) | 10–20 years (n = 5) | 10–20 years (n = 4) |
| | >20 years (n = 1) | >20 years | >20 years (n = 2) |
| | Don't know (n = 2) | | |
| Level of seniority (self-identified) | *Unknown–anonymous survey* | Senior researcher (n = 9) | Senior researcher (n = 11) |
| | | Early Career Researcher (n = 2) | |
| Total | N = 22–41* | | |

* Given the survey was anonymous we do not know how much overlap there was between survey participants, key informants and workshop participants therefore we are providing a range of total participants.

mechanisms by which they categorise and label countries, so it is difficult to achieve consensus on categorisation even within terms. Categorising countries by their geographical location, or economic status in either way was viewed to be inaccurate, but inaccurate geography was viewed more favourably. The term 'majority world' was also offered as an alternative in the workshop.

'*Again, I think the LMICs for me is not palatable. . . it's just a label. . .it's not realistic. [We] might be a poor nation, but now we have got millionaires. . .'* (KII–senior, man, Southern Africa)

## Fair opportunity—Funding streams

While the majority of survey respondents (17 out of 21) agreed that the partnerships with LSTM-Liverpool attempt to ensure that research funder demands do not cause unfairness in partnership, three respondents disagreed. This may be reflective of the funding structures that give LSTM-Liverpool more power within the partnership that emerged in KIIs. Partners reported that most projects in the partnerships were funded from sources within the UK. This was described as shaping the power dynamics in the partnership, mainly attributed to LSTM-Liverpool's interaction with funders:

'*[LSTM] are the ones who deal with the donors, because that's just how our funding is arranged. So, there will always be more power for the people at LSTM, in terms of taking charge of the project. . . And all the projects that we've done, they have always been prime*' (KII–Senior, Man, Southern Africa)

Budget allocations were described as being discussed collaboratively and transparently. Partners reported LSTM-Liverpool supported with financial management capacity, technical and liaison support when engaging with funders. All partners were affected by the UK's 2021 Overseas Development Assistance (ODA) funding cuts following COVID-19. Partners reported having open and transparent discussion on how to work with limited funds after the funding cuts and reported positive outcomes of LSTM-Liverpool Principal Investigators (PIs) negotiating down the size of the cuts. Unsurprisingly, funding structures that rely on Northern funded research were described as stifling global equity; domestic research budgets were described as critical to equity:

'*. . .So, in terms of decision making, the global South has been included in decision making in many ways. But we might be disadvantaged, or we might not reach the absolute end of equity, if we are not financially independent. So, I think that if we really want to maintain independence, then countries in the global South should begin to think of how they can generate research funding from within. . . So, I think in terms of resource capacity, that is one area that might likely stifle global equity.*' (KII- Senior, Man, East Africa)

It was suggested that LSTM-Liverpool can play a role in facilitating dialogue between UK funders and Southern partners to foster transparency and build the capacities of both funders to understand contextual realities in the global South, and of partners in understanding funding mechanisms. Further, through our participatory workshop, it was highlighted that the space to bring together different partners who do not normally come together was valuable and something LSTM-Liverpool could continue to facilitate to support fair opportunity.

## Fair opportunity—Agenda setting

Agenda setting between LSTM-Liverpool and partners was often described as being defined mutually and often a participatory process, particularly when physically together. Proposal writing workshops and early engagement with partners was seen to help implementation and sustainability. Partners mentioned being involved early in the partnership(s) and able to input into the aims, objectives, methods for the partnership proposal and co-develop the Theory of Change to reflect contextual realities. Early inclusion is fundamental to equitable partnerships and is therefore reflected as principle one (Box 1).

'*LSTM is really good at ensuring that its partners are engaged as early possible on any opportunity that arises. This has led to not only smooth implementation of the project(s), but implementing very successful projects and partnerships where each partner is satisfied and even wishing for more*'. (Survey respondent, man, Southern Africa)

Agenda setting was also mentioned as being defined by the funding call and the global funding architecture, which is predicated on global power dynamics. Ensuring this power dynamic is then not replicated within the partnership itself was highlighted by one participant.

'*Somehow our partnership is also influenced by the global agenda setting process. . . our partners in higher income countries who dominate in the selling process. However, I'm not blaming, but that has some kind of synergies or linkages between the partnership of two institutions like [organisation] and LSTM team. However, understanding of the global agenda setting process, not repeating the weaknesses of that process, while agenda setting between two partners is very, very important. . . open communication really that's the first thing . . .and not dominating the discussion is very important.*' (KII-Senior, Man, Southern Asia)

'*Uh, and I think that's driven to a large extent by the fact that high income countries are putting in more money and therefore they are answerable to their parliaments and not [our] parliament. I don't think we will get into a situation where that is completely balanced out. I'm not even sure that it needs to be completely balanced out.*' (KII- Senior, Man, East Africa)

Agenda setting in response to local needs was reported to be a balance of country priorities and funder influences. It was noted that the UK funding landscape is shifting to allow a broader focus, offering more space to align with country priorities. Long term engagement and familiarity with contexts means that partners can respond to local needs in priority setting. One participant who had a longer-term engagement with LSTM also noted that the response to local priorities had been given more weight in recent years, whereas historically LSTM's priorities had been dominant. Despite funding streams generally flowing from North to South, there were examples where Southern organisations were the consortium lead and funding recipient.

'*We do the agenda jointly. I mean, [consortium name] is leading the research uptake, the expectation from the team is that Africa leads that programme, leads the consortium, but, leading doesn't mean that you're determining everything, you basically co-create within the consortium. But we lead those components, and so the agenda is set by the consortium itself.*' (KII-Senior, man, East Africa)

Space for innovation within protocol and grant writing was described as a current gap as many partnerships are focused on executing research via traditional methods. This raises the

question: *what spaces can be created for co-production and co-creation through prioritising local knowledge*? The importance of relevance to the research agenda is reflected in principle three (Box 1).

### Fair process—Collaboration with LSTM—Liverpool

Collaboration between partners and LSTM-Liverpool took different forms, namely research, capacity strengthening, teaching, research uptake, advocacy and publication. Length of engagement and physical proximity were noted to influence the collaboration. One respondent noted the benefit of LSTM-Liverpool having a physical in-country presence when collaborating, as opposed to only attending meetings. Physical separation was reported to negatively impact collaboration as institutions are perceived to not "*walk the talk*" (KII- senior, man, Southern Africa). Two-way university faculty exchanges were also stated as a concrete way to strengthen the partnership.

The personality of PIs from LSTM-Liverpool were also influential on the nature of the collaboration. Some PIs were mentioned for their excellence and commitment to equity in partnerships, which influenced a valued, personal relationship with LSTM-Liverpool projects. How much this is reflective of a culture of fairness within LSTM-Liverpool was not clear. Multi-directionality was also stated to be an important factor in the nature of the collaboration.

'*What I like about* [consortium] *what's been good, we had [name of PIs] and [PI name]'s approach is very different. . . So you know, in some cases, . . ..there is engagement, you do great research, but there isn't that feeling of you're part of a community. But with [consortium] I feel like we're part of a community*'(KII—senior, woman, South Asia)

'*Actually, we have a rotating chair, I think way different partners take charge in chairing the management meeting. This shows that there are efforts to make sure that partners especially in the south involvement in the project.*' (KII-Early Career Researcher (ECR), man East Africa)

Decision making within partnerships was described by most participants as fair and respectful. Nevertheless, two survey respondents felt power dynamics that influence decision making, are not always explicitly discussed and roles are not always clearly defined. Developing organograms and advocating for rotating chairs in meetings emerged from KIIs as valuable to understand who will lead what and share power at the outset. The importance of transparency across partnerships has been reflected in principle two (Box 1).

It was also noted that LSTM-Liverpool has played a role in establishing transboundary organisations. While LSTM-Liverpool's supportive role was acknowledged, the analogy of LSTM-Liverpool as a parent and the partner's organisation as the child was used to describe the relationship, suggesting a paternalistic collaboration with links tocoloniality. The analogy was also used to state the need for independence, suggesting how although partnerships, capacities and relationships evolve through time, power dynamics can be more entrenched.

"*I think* [of LSTM] *as our parent, we take ourselves as, as the brainchild of the LSTM. . . over the years, we've seen LSTM supporting staff from* [partner name] *with education opportunities, master's degrees, PhDs and I happen to be one of those are beneficiaries of this partnership. . . So they've seen us through thick and thin. And that's why I use the term—like a baby, LSTM as a parent. . .What LSTM is trying to achieve should actually also be seen in practice. . .you know, there reaches a point whereby a child is grown and wants to walk alone.*

*And I know that's very hard for the parent to let go. But sometimes you do have to let go and just watch from afar. So maybe that's where LSTM and [organisation] are."* (KII- senior, man, Southern Africa).

To support equity in fair process a suggestion was to leverage virtual platforms for knowledge exchange and networking to expand and develop collaborations, as these can enhance equity as costly travel and visa complications are avoided.

## Fair benefit sharing—Capacity strengthening

Capacity strengthening was seen as a strength of LSTM-Liverpool's engagement. Its importance was emphasised by both senior researchers and ECRs, and as such it has been reflected in principle four (Box 1). It occurred intentionally and evolved naturally by nature of the partnership. There was value placed on data sharing, post research agendas, documentation, using new and innovative methods and developing skills in communicating research. Project length was described as an important factor in influencing the success of capacity strengthening initiatives, with longer projects, for example five years, allowing space for individual capacities to be enhanced.

'. . .*projects that are two years you just getting into the momentum. And then it's like see you later, goodbye, wrap up. . . But is that when you have a longer-term project, I mean, colleagues come and go, but the few that [stay], you can see them grow in terms of their skills. And I think that's very good.*' (KII—senior, woman, South Asia)

Capacity strengthening was mostly focused on an individual level particularly from the global South, with MScs, PhDs and post-docs being highlighted as particularly beneficial for mid-career researchers. Careful attention needs to be paid to the distribution of academic opportunities within consortium partnerships to support equity:

'*Training needs to be equally considered in participating countries for example all PhD opportunities went to two countries while we had more several participating countries. A selection criteria need to be purposeful as every country needs to strengthen research capacity*' (Survey respondent, East Africa)

It was noted that projects with a transparent and horizontal hierarchy structure have benefits to individual capacity strengthening–particularly for early and mid-career researchers. Individual capacity strengthening also evolved from community engagement—'unlearning academia'—these projects were reported to develop skills in blog writing, photo narrative writing and podcasts.

'*Capacity strengthening has really been at its best since we started collaborating with LSTM. We have our capacity building in different areas relating to research, to communication, to research methods, . . . it has been a wide range of opportunities for capacity building and it has really enhanced our work, even in different projects that are not related. . .*' (KII- ECR, man, West Africa)

Funding for institutional capacity building was identified as an unmet need but was reported as critical for establishing credibility for organisations. One example was mentioned whereby staff were sent to do secondments at LSTM-Liverpool and returned to strengthen institutional capacity at home. Critically however, it was noted that this should be multi-

directional, which is why the term capacity strengthening and exchange is reflected in principle four, which also includes a specific focus on non-academic staff.

While some participants mentioned improvements in budgeting and financial management because of the partnership, many partners mentioned the need for capacity strengthening of financial management systems, research and management capacity as well as publication for high impact journals. Survey results suggested that there are gaps in the provision of resources for capacity strengthening of project staff and it was also acknowledged that administrative and project management staff are often overlooked. This is a critical area for consideration as one participant emphasised the need to acknowledge transboundary partner institutional and contextual limitations, which may impede on opportunities to lead partnerships and deliver to specific templates for reporting.

'*We are an organisation established and evolving and struggling in developing world with lots of limitations . . . resource limitation, limitation in terms of capacity, limitation in terms of the institutional component where you require investing quite a lot in terms of building institution, not necessarily that is up to the standard of the LSTM institution. . . . But using a standard global templates that are the template for all one size fit for all, while setting the standards from the LSTM side. . . that presents limitation to organisations like us, we are not as big as LSTM, but big does not mean that it should dominate the standards. . .I think it's important that there is a rigid trust built in both end in terms of managing a project under this partisan framework. And understandably, it takes time to build that trust. . . in order to develop trust both end by knowing the institutional practice. . ..*' (KII senior, man, South Asia)

Recommendations included completing a competency assessment to establish capacity strengthening needs in research partnerships and to onboard all partners with established and clear templates for planning, budgeting and reporting. One participant suggested that LSTM-Liverpool could support partners with access to academic and online training courses currently available only to LSTM-Liverpool staff and students e.g. in safeguarding, and access to libraries. It was also emphasised a focus on strengthening capacity of partners to innovate and think outside of the traditional research process would be beneficial.

'*So I think it's the capacity to understand where innovations can be. . .there are small things such as theory of change, etc. These are things which are not normal in our traditional medical research. So it's something we're also taking on board, it's more of a global health issue, public health. So these are things we're learning as we move along. So.., the UK may be way ahead of us, but we're learning so even when we write proposals, we always gain knowledge. They understand how things work on the ground, what the problems are, but we also gain that understanding of how to write certain proposals.*' (KII–senior, woman, East Africa)

### Fair benefit sharing—Impacts and outputs

Research uptake at the local and national level is reported to be led by partners, whereas LSTM-Liverpool was reported to facilitate at global level—including publications. The focus on LSTM-Liverpool's impact at the global level reinforces broader power relations in knowledge generation within global health and does need to be re-evaluated to support equitable partnerships. Nonetheless, partners' strong relations with the Ministries of Health were stated to be valuable in policy impact and raising visibility. One partnerstated their existing strong relationships with policy makers at national and local level should not be undervalued in

creating policy change. Another spoke to how the strengths of each institution come together for impact.

Authorship with LSTM-Liverpool was described as equitable and commendable. Good practice involved early discussions on equitable authorship, a commitment to follow journal protocols and guidelines and utilising opportunities for joint first and last authorship positions. It was noted that the publication process is usually led by LSTM-Liverpool and Northern partners, and that this is also shaped by funding, limiting equitable access to publishing. "*To be honest, I can't afford $5000 to publish in maybe in BMJ [British Medical Journal] or so*" (KII -Senior, man, West Africa). It was felt generally, academics put more focus on outputs and publications while non-academic partners understand the local context and influence policy in different ways other than publication, but this is not always considered as being of equal value.

A critical area of concern, however, with regards to impact and uptake is insensitive and outdated legal language in contracts with LSTM-Liverpool–that often state that data is owned by LSTM-Liverpool. This was described as promoting lack of ownership, and limits opportunities for Southern led authorship and demands urgent attention across Northern institutions.

> '*Legal language in contract that data is owned by LSTM, data should be owned by researchers with right to publish. Promotes lack of ownership.*' (Survey respondent, Woman, East Africa)

This highlights the need for data sharing agreements to be put in place which give partners the power and rights to their own data and grant LSTM-Liverpool access to support in dissemination and power sharing (reflected in principle two–Box 1). Further suggestions to increase equity in fair benefit sharing was for LSTM-Liverpool to leverage their position to increase power of transboundary partners by introducing and nominating partners to be involved in international forums such as technical working groups or conferences, or by promoting transboundary partners in publications. This is reflected in principle five (Box 1).

## Co-developed principles for equitable partnership

The values that survey and interview participants identified as critical to equitable partnerships are depicted in Fig 2. These values were presented in the participatory workshop, alongside

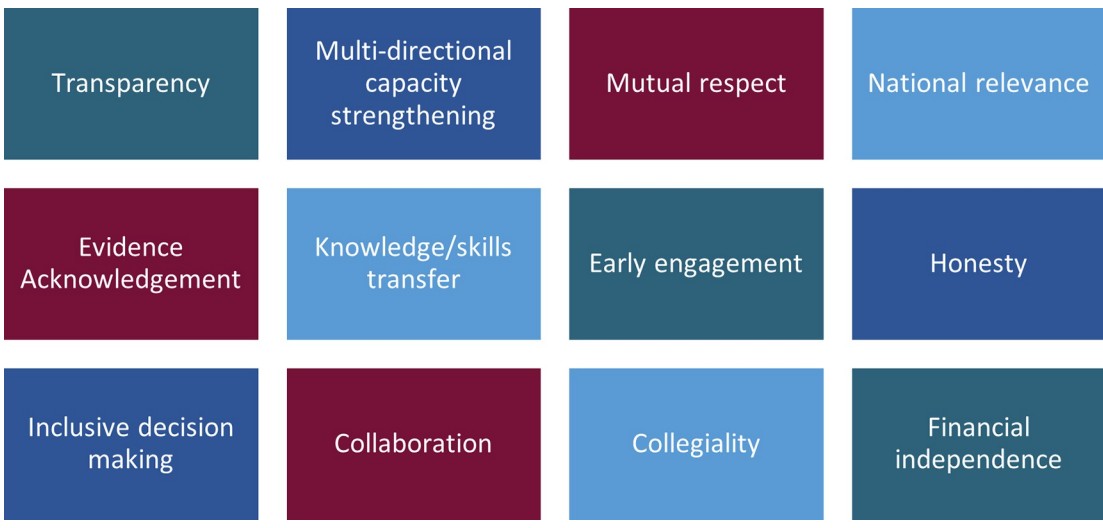

**Fig 2. Research derived values of equitable partnerships.**

draft principles that had been developed from these values (RS, RK, SC). Most workshop respondents agreed that their partnerships with LSTM-Liverpool matched these values. LSTM-Liverpool was regarded as doing well in equitable partnerships through early collaboration and engagement, support for staff, honest and transparent processes and a focus on co-leadership. Partners within the workshop further edited and refined the draft principles. Key edits pertained to capacity strengthening being multi-directional (reflected in principle six–Box 1), and the inclusion of values of the TRUST Global Code of Conduct (reflected in principle seven–Box 1) [34]. These principles were then shared again and further refined. Final principles are shown in Box 1.

In the workshop, many partners suggested that equity could be evaluated against agreed parameters, through follow up interviews over time, recorded documents, regular feedback and multi-partner project evaluation records. It was noted that accountability mechanisms for voicing partnership concerns exist: through open communication (particularly when partnerships have been built over the years); regular management meetings; safeguarding protocols and working groups. It is important to note however, that this was stated by respondents who work at senior levels.

## Discussion

This study is the first time LSTM-Liverpool's partners have been actively engaged to co-develop institutional principles to support meaningful and equitable engagement. We sought to foreground Southern perspectives, languages and cultures in progressing equitable partnerships and dismantling the 'master's house' of global health [27, 31, 35]. This is a critical starting point towards more transformative partnerships, but we recognise the need to ensure this process is inclusive and representative of diverse knowledge that extends beyond our existing partners and power structures. As Mogaka et al. (2021) note, global South participation in the global health decolonisation efforts is limited to researchers who have emerged through the current colonial global health structure [36].

### Our findings

On fair opportunity there were positive accounts of partners being involved in agenda setting from the outset—able to shape and input into the design of research projects and adapt theories of change to reflect contextual realities. Fair process revealed that the collaborations were viewed as multi-directional, positive and personal. Causes for concern around funder requirements creating unfairness within the partnership and outdated language in contracting and partnerships not supporting equity were also raised in the anonymous survey responses and require further exploration and action from LSTM-Liverpool. The need for funders and donors, primarily from high-income countries, to redefine their roles in shaping equity by centering and directly supporting Southern researchers and institutions is also highlighted by Charani *et al*. [3]. There was an emphasis on capacity strengthening in LSTM-Liverpool's partnerships that was valued by partners, though, it was highlighted this needs to be extended to research support staff and professional services, which was reflected in principle four (Box 1). Finally, fair benefit sharing revealed equitable processes around authorship supported by upfront discussions and policy influence. These are reflected in principles one, two and three which speak to early, relevant and transparent engagement.

The interlinkages between the three domains, and how they are situated within broader political economy of research partnerships clearly emerged in our findings. For example, funding structures underpin power structures that dictate fair opportunities, both in research grants and shaping the research agenda. Funding also influenced ability to travel, network and

share findings, which can shape career trajectories. This was important for fair benefit sharing but also fair opportunity as it was reported collaborations sometimes started through networking at international conferences. Funding also influences fair benefit sharing as high paywalls limit both sharing of research findings and access to research findings for many Southern institutions.

Findings from some KIIs had clear links to coloniality, for example the language of LSTM being the parent, speaks to paternalistic collaborations which Okeke has dubbed '*the little brother effect*' and requires LSTM-Liverpool staff, as well as staff in other Northern institutions, to be critically reflexive on their well-meaning role as 'older brother' [37] and subtle paternalism [38]. Colonial assumptions about knowledge superiority were also discussed where partners described how capacity strengthening opportunities usually flowed from North to South. Southern staff would go to do secondments at LSTM-Liverpool and other Northern institutions, but it was less expected that Northern staff would come to learn at Southern institutions, which speaks to entrenched knowledge hierarchies. As Abimbola reflects, "*We mustn't give in to the notion that the global South must come to the global North's table to be seen as knowers*" [39]. It was also noted that capacity strengthening occurred from unlearning academia and learning from communities, yet the spaces for innovative practices and inclusion of local knowledge through co-produced proposal development were more limited. This highlights the importance of ensuring funding and action to support multi-directional capacity strengthening and exchange initiatives for equitable partnerships, which was stressed and is reflected in co-constructed principles (for example, principles four, five and six) [21].

There was also debate around accepted terminology for geographical and political areas. What is clear however, is that efforts to homogenise large swathes of geographies into clear and widely accepted terminologies fails to capture the nuances and the varying facets that characterise privilege and disadvantage in those settings. Instead, we should embrace 'pluriversality' and stop trying to reach consensus, which has been argued a necessity for decolonisation [40]. Language however, is not the sole issue, rather we need to think critically about the multiple power relationships that language represents and that tailoring language may be appropriate [12, 41]. The process of engaging in reflection, dialogue and debate is one of the 'small wins' we can make on the journey towards rebalancing existing power dynamics [11, 25].

## Principles

The process taken to arrive at the seven actionable co-developed principles to guide equitable research collaborations (Box 1) took various methods and iterative consulting (Fig 1). These principles reflect some of the key issues that are centre stage in other definitions or approaches in recent work (for example there are some similarities in the Essence in Health Research and UK Collaborative on Development Research (UKCDR) best practice guide to equitable partnerships with a focus on mutual participation, trust and respect) [42]. While these findings are not novel, this study further generates additional supporting evidence on the importance of these themes with new nuances [2, 21, 34, 42–44]. Critically, these principles were developed with transboundary partners, which has been found to be lacking in existing principles for equitable partnership [2, 27]. The process itself was also of value. LSTM-Liverpool and partners took time to deliberately reflect and define the partnership, a process that should be maintained regularly. Principles will be subject to ongoing review, including reflection on who has inputted, working to address the geographical and epistemic gaps and include critical reflection on who are partners are, and who benefits–this entails reorienting research to ensure it is reflective of values, cultures, structures, roles and resources of indigenous cultures in the Global South [35]

These principles must not be viewed as operating in a vacuum. We, as researchers need to engage with the research eco-system that shapes the partnership, creating space for mutual learning and adapt to changes in the external context [23]. For example, partnerships are influenced by the changing global funding architecture; the recent climate of UK ODA funding cuts amidst a global pandemic brought unexpected budget cuts and additional challenges to transboundary research, as highlighted by respondents, and in recent literature on research partnerships [45]. Individual and institutional action is critical but also needs to be part of a wider process of change within the broader political and funding ecosystems that shape and underpin research partnerships and maintain epistemic injustice through funding infrastructure (donors) and knowledge dissemination avenues (journals). Indeed, Voller *et al* (2022) note how guidelines may not always fully acknowledge the structural barriers and competing interests that get in the way of changes being realised [2].

We encourage other institutions to engage in a similar reflective process with partners. Acknowledging the burden on time in co-creating these with transboundary partners, we welcome use of these principles for adaptation.

## What the implications of our findings–for LSTM-Liverpool and partners; and for broader research infrastructure

**Role of funders and the future of funding.** Funding structures that typically flow from North to South were lamented by our respondents as inhibiting equitable partnerships more generally. Funders are required to engage with, and embrace, the complexity of equitable research partnerships and contextually specific research to advance health systems research [43]. This requires a shift from Eurocentric modalities of research funding to support Southern led research and co-production approaches for sustained engagement, which is beginning to occur [24, 46]. Some funders are now funding Southern institutions directly, as well as setting up development funds to foster and enable partnership working in the grant development process (e.g. NIHR, UKCDR), which is recognised as a start. Indeed, this is also a recommendation in UKCDR's recent best practice document on supporting equitable partnerships [42]. However, funders need to pay critical attention to ensuring that funding provision is equitable and that by funding Southern institutions directly, they do not leave them worse off by imposing excessive rules, regulations and costs that the organisations may not be able to bear. As highlighted by respondents, institutional capacity is not always as strong as large academic institutions and as such there needs to be flexibility in reporting requirements. Funds for embedded operational capacity strengthening and ongoing work to distil learnings on equitable partnership are also critical–these must be co-created with, or led by Southern institutions, so it has value for them, is appropriately embedded in contexts, and not just based on the funders needs. The benefits of long-term funding cycles on capacity strengthening were also highlighted by respondents and should be noted by funders, as capacity strengthening has been shown to be critical to fairness in research partnerships [21].

As described by one of our respondents, investment into locally led research is also critical. Domestic sources of funding for research are critical to mitigate against power imbalances in global health research and should be what we strive for. The political climate of many Northern countries may even force this–recent UK ODA funding cuts were a source of concern for many partners we spoke with and risked undermining relationships with UK institutions. Further, the UK and Italy were notably absent from commitments to the Global Fund in September, 2022 who missed their replenishment targets [47]. Despite their absence, many Southern countries did pledge, some such as Malawi, for the first time, highlighting a shift in who is funding global health.

**Knowledge generation and dissemination: Role of journals, conferences and beyond.** Knowledge systems in research and academia need critical review to address epistemic injustice [23, 39, 48]. Equitable partnerships not only encompass mutually beneficial processes, but also products [49]. There is a growing body of work on research equity, authorship, and the role of academic journals in the context of international health research partnerships—see for example the "Consensus statement on measures to promote equitable authorship in research publication from international research partnerships" [50]. Several journals (for example Health Policy and Planning and PLOS Global Health) are now including reflexivity or inclusivity statements, which require author teams to critically reflect on roles and responsibilities and demonstrate inclusivity by regional location (S4 Consenus statement). Equity in authorship and in access to papers has also been a key theme in recent conferences [51]. Hosting conferences and forums virtually, or in hybrid form, may also confer a benefit in terms of equity of participation when compared to in person meetings that may be expensive and include visa complications (passport privileges emerging as an important area for consideration). Practical suggestions to support equity from our findings were having authorship discussions up front and making use of joint first and last author positions, as we have done in this paper. Transparency and trusting relationships between partners are therefore critical to facilitate these potentially uncomfortable discussions and is included in our second principle (Box 1).

Inequities in knowledge dissemination and sharing remain. Our research participants highlighted barriers in accessing resources to publish, and challenges in accessing resources that are not open access. There also inequities in who publishes: a recent review of papers reporting research trials from the global South showed from 1990–2013 papers with a first author from the global South increased 2.8 times while papers with a first author from the global North increased 11.8 [52, 53]. To date, knowledge generation has favoured those in the North over the South [52], evidenced by Africa-only publications having lower impact factors than internationally co-authored work [24, 54]. This is maintained by the so called 'international' reputable academic journals such as The Lancet or British Medical Journal that are often headquartered in the global North, as raised in our external validation exercise [50]. To truly rebalance scientific knowledge systems we should strive towards knowledge that is created and published within the global South as a valuable pathway for Southern led and globally disseminated knowledge. This calls for an upheaval to the dominance of Western languages in scientific knowledge production and embracing indigenous languages and cultures and other ways of knowing [24]. It was also highlighted how equal value should be given to understanding the context and policy space as well as academic publications, speaking to the need for structural change in how we value knowledge in global health. This links to Aubel and Dixon (2022) who argue that global health reflects Euro-centric values and overlook collectivist cultures and perspectives [35]. Transformative research collaborations require reframing of research and practice to reflect structure, roles and resources of indigenous cultures in the Global South.

**What are the implications for LSTM and next steps?.** The measure of an equitable collaboration was sometimes measured by the merits and approach of individual PIs rather than as the culture of LSTM-Liverpool as a whole. A key point that emerged from the respective discussions is '*how will the principles be rolled out and assessed*? *Whose responsibility is it to review adherence to the guidelines and document learning to guide future work*?' The principles that were co-created are intended to act as a means for accountability. They should be agreed upon at the beginning of partnerships and reviewed throughout. These have to be lived by LSTM staff and transboundary partners and can hopefully be leveraged by partners to use in external relationships too. To ensure they do not simply become a 'tick box exercise' it is critical to have accountability mechanisms in place.

LSTM's current actions to work towards these principles include: 1) the establishment of a partnership council with LSTM's hub (A Global Hub refers to a Transboundary Partnership through which an independent organisation (or group of organisations) based outside of the United Kingdom has an established association with LSTM characterised by ability or potential to deliver to an international standard on all domains of Academic endeavour: Research (grant income and publications), Education (postgraduate teaching and postgraduate supervision), and Knowledge Exchange. Current global hubs with LSTM-Liverpool include Malawi-Liverpool-Wellcome Programme, Malawi; KEMRI Kisumu,Kenya and CeSHHAR, Zimbabwe) partners (principle six). Seeking joint accountability to the seven principles is a core objective in the council terms of reference which have been agreed by all hub and LSTM senior management teams. The bi-monthly council meetings are chaired in rotation by each of the organisations. The potential to open some future meetings to wider participation is under discussion, recognising LSTM's convening power and the value of coming together to share ideas outside of the confines of existing research partnerships, that was noted in the workshop.; 2) a requirement to report on measurable equity criteria and reflexive questions within an internal research database (principles one-four, seven); 3) ongoing training on embedding equity within programmes that builds on this research; 4) examination into the processes of ethics, safeguarding and research integrity and governance; and 5) an overhaul of outdated contracts [55]. LSTM's next steps involve fostering a culture among all staff (legal, financial, administration, researchers, lecturers etc.) that supports and values approaches to supporting equitable partnerships. This will require relative power sharing, both in terms of securing funding but also in promotion of Southern colleagues for benefit sharing where appropriate. It requires a re-evaluation of academic currency and promotion criteria to value the soft skills of partnerships as well as the measurable outputs from it [5]. A tension that Northern researchers face is the balance between embracing equitable partnerships in a way that is not detrimental to the institution, or individual's careers. This is particularly critical, as without attention to this we risk creating hierarchies where those who get promoted to senior positions continue to perpetuate a culture of inequity. It may also threaten the applicability of principle five, which relies on using a position of power for the benefit of transboundary partners where appropriate. How LSTM-Liverpool chooses to 'measure' interpersonal relationships of staff is important to support accountability and value those who cede power and create space for partners. Such measures may include partnership surveys or seeking feedback from transboundary partners as a routine part of academic promotion criteria.

Reflexive practice and action are critical to moving beyond equitable partnerships towards transformative partnerships. Staff should engage in continuous, critical intellectual exchange and learning to inform structural shifts towards transformative collaborations [24]. Further, LSTM and academic institutions may wish to think about who has access to short courses and resources–whether this is limited to current students, or extended to alumni and transboundary partners and community co-researchers; how they engage in faculty exchanges and whether Southern experts are called upon to strengthen LSTM's capacity as envisioned by Abimbola and Pai (2020) [56]. It will also be crucial for staff in LSTM-Liverpool to critically reflect on their own positionality and how to embed transformative learning in curricula to sensitise upcoming generations of staff, and support fair and transparent processes that acknowledge different people's multiple roles and responsibilities, possibly adopting the use of a praxis cycle, which has been applied to educational practices [57]. Finally, on benefit sharing it was noted by respondents that less value was given to non-academic ways of policy uptake influence, as opposed to publications favoured by academic partners and current academic reward structures. This requires critical reflection on how we value policy change, and who it serves to benefit. Shouldn't the policy and practice impact to communities hold greater value

than the publication that inputted into it? Yet, eschewing academic rewards has knock on disparities in career opportunities that accrue through publication leadership in criteria for academic promotion and further grant capture [24].

## Limitations

We began the critical process of engaging with partners to co-develop principles for equitable partnerships, though there were limitations to our approach. Firstly, we sampled from our existing partners. This may have introduced some bias as partners generally felt positive about the relationship. Though we sought to recruit diversity in partners, it is likely that those who had a positive view of LSTM-Liverpool opted into the study more so than those who felt indifferently, or negatively. Given recruitment processes were initiated through LSTM-Liverpool, participants may have felt they needed to respond positively about the institution, we aimed to mitigate this through interviews being undertaken by LVCT Health (RK) and (RS), who was not working for LSTM at the time of study. We also mostly recruited partners who had a senior role within their organisations for the KIIs and the workshop. Although we intended to engage partners from a range of career stages, the ethical approval process to conduct the study required the approval of the organisation head before we could approach others. Senior respondents were asked to cascade the recruitment call within their organisations but this relied on their active engagement and likely limited our respondent pool. There is a need to engage further with more junior researchers from transboundary partners to mitigate against power imbalances that may have limited their participation. Further, professional services staff were not included in our KIIs or co-development workshop which is a clear limitation and area for future research. Again, the ethics procedure for recruitment meant that hierarchies of knowledge within research institutions were reflected in our respondent pool. We also had a small sample of respondents though the multi-method approach (Fig 1) aimed to mitigate this by capturing a range of diverse views and perspectives through time, providing a holistic picture of the nature of partnerships.

The challenges of recruitment also impacted the global diversity and range of organisations that we spoke with, and who was at the table in this analytical process, for example there were no partners in Central or South America included or Eastern Europe. While we had a respondent from the Middle East in the participatory workshop who helped shape the principles, we did not have respondents from this region in our survey or KIIs. These regional exclusions are important- our sample included 20 organisations reflective of the areas in which LSTM-Liverpool works. This highlights another contention, in so far as who LSTM has as partners may not be equitable before the partnership has begun. Indeed as Müller notes–countries within eastern Europe fall into an epistemic space they term 'the global east'–these countries represent a dual exclusion as they are neither considered the global north or south [58]. LSTM's partnerships possibly reflect this exclusion from the global view.

Many organisations who we spoke with were engaged with only a handful of PIs, so to what extent our findings reflect working partnerships across different departments within LSTM is also uncertain. Finally, our research was conducted at a particular moment in time, COVID-19 had led to UK ODA funding cuts and high levels of uncertainty and precarity in relationships, with early and mid-career researchers impacted most of all. This allowed us to explore how LSTM responds to challenges, but our results need to be interpreted with this context in mind. Despite these limitations, we feel that the participatory analysis, co-construction and stakeholder checking processes aimed to bring diverse views and perspectives into the process and to mitigate against some of the recruitment and power issues that may have influenced the research process.

## Conclusion

Fostering equitable partnerships means we must confront uncomfortable truths within partnerships; acknowledging and discussing how funding sources, donor priorities, history and language shape existing power asymmetries, as well as critical reflections on who partners are. LSTM-Liverpool is conducting an ongoing process of deep reflection with transboundary partners. It is important to note that equitable partnerships are not something to be 'achieved' rather, they demand ongoing reflection and dialogue, seeking to do better and to dismantle harmful power structures that perpetuate coloniality within global health. These principles include the perspectives of transboundary partners and represent small steps on this journey towards broader transformation and a rebalancing of power. We hope these principles will be helpful to others and will be adopted more widely by LSTM-Liverpool, and transboundary partners to support accountability as they strengthen existing relationships and form new ones.

## Supporting information

**S1 Checklist. Inclusivity in global research questionnaire.**
(DOCX)

**S1 Text. LSTM partnership survey– 2021.**
(DOCX)

**S2 Text. Topic guide for key informant interviews.**
(DOCX)

**S3 Text. Consensus statement.**
(DOCX)

## Acknowledgments

We would like to acknowledge all the partners that gave their time and thoughtful insight to shape this research and co-develop the principles presented. We thank Professor Frances Cowan and Dr. Webster Mavhu from LSTM and the Centre for Sexual and HIV and AIDS Research, Zimbabwe (CeSHHAR), Dr. Ndekya M. Oriyo, National Institute for Medical Research (NIMR) and Professor Asma El Sony (Epi-Lab) for their critical comments and for the other valued, anonymous inputs we received. Thank you for administrative support from Annmarie Hand and ADAPT at LSTM: Beth Hollihead, Faye Moody and Tracy Owen.

## Author Contributions

**Conceptualization:** Robinson Karuga, Rosie Steege, Bertie Squire, Sally Theobald, Lilian Otiso.

**Data curation:** Robinson Karuga, Rosie Steege, Shahreen Chowdhury.

**Formal analysis:** Robinson Karuga, Rosie Steege, Shahreen Chowdhury.

**Funding acquisition:** Bertie Squire, Sally Theobald.

**Methodology:** Robinson Karuga, Rosie Steege, Shahreen Chowdhury, Sally Theobald, Lilian Otiso.

**Validation:** Robinson Karuga, Rosie Steege, Bertie Squire, Sally Theobald, Lilian Otiso.

**Writing – original draft:** Robinson Karuga, Rosie Steege, Shahreen Chowdhury.

**Writing – review & editing:** Robinson Karuga, Rosie Steege, Shahreen Chowdhury, Bertie Squire, Sally Theobald, Lilian Otiso.

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
