## [Decision Letter · Decision Letter 0]

23 Aug 2023

PGPH-D-23-00801

A multi-step analysis and co-produced principles to support Equitable Partnership with Liverpool School of Tropical Medicine, 125 years on

Dear Dr. Chowdhury,

Thank you for submitting your manuscript to PLOS Global Public Health. After careful consideration, we feel that it has merit but does not fully meet PLOS Global Public Health’s publication criteria as it currently stands. Therefore, we invite you to submit a revised version of the manuscript that addresses the points raised during the review process.

This is a very important topic and you have done a significant amount of work towards developing your co-produced principles. I believe that this paper has promise, but will need some significant revisions.

First, there are some methodological issues where further explanation would be more useful. Also, the paper could be organized more effectively, with more signposting and reference to the principles earlier on. Currently different sections in the paper could be make to flow together more coherently.

In addition, here are some specific comments to provide you more detail on the methodological questions I had:

1. Overall, showing more of the data at different phases could be helpful. For example, you could provide the table of partner organizations mentioned in Line 127.

2. I would also discuss further limitations from the various samples/recruitment phases used. For example, Would would only working with willing senior leads at LSTM affect the conclusions/potential guidelines that could be included?

3. The Middle East is mentioned as a region early on (and between the Middle East and North Africa interviews/survey respondents, you do have a few data points here), but then the paper only mentions Africa and Asia. I was wondering why this is the case. It would strengthen the piece to discuss your data for that region as well since it was included in the analysis.

4. For the workshop discussed in Line 155, how many participants were from LMICs? It could help to show the balance between representation from HICs and LMICs through all of the stages of recruitment.

5. Line 167 discusses the validation exercises, but I do not see a serious discussion and analysis of these inputs.

I would also encourage you to address the reviewers comments.

We look forward to receiving your revised manuscript.

Kind regards,

Renu Singh

Guest Editor

Journal Requirements:

Reviewers' comments:

Reviewer's Responses to Questions

**Comments to the Author**

1. Does this manuscript meet PLOS Global Public Health’s publication criteria? Is the manuscript technically sound, and do the data support the conclusions? The manuscript must describe methodologically and ethically rigorous research with conclusions that are appropriately drawn based on the data presented.

Reviewer #1: Yes

2. Has the statistical analysis been performed appropriately and rigorously?

Reviewer #1: N/A

3. Have the authors made all data underlying the findings in their manuscript fully available (please refer to the Data Availability Statement at the start of the manuscript PDF file)?

Reviewer #1: No

4. Is the manuscript presented in an intelligible fashion and written in standard English?

Reviewer #1: Yes

5. Review Comments to the Author

Reviewer #1: Overall, this is a well-written paper that clearly describes a well-designed and implemented piece of research. The findings and arising discussion points are not novel, but nevertheless contribute to the body of literature on equitable partnership principles and practice by reinforcing themes described elsewhere. Greater reference in the discussion section in particular to the wider literature, including recently published guidelines, would further strengthen the manuscript.

Detailed comments:

Line 17-18. Confusing to the reader to mention the partners invited to participate in the survey and that the survey was conducted ‘in partnership with LSTM-Liverpool’. Could it instead read ‘the online survey was conducted by LVCT Health and LSTM-Liverpool’ or simply delete this part of the sentence and focus on the respondents whom the survey was aimed at.

Row 41 or 58 Suggest referring to the KFPE Principles (3rd ed) which use the term ‘transboundary’ when introducing the use of this of term in this paper.

Row 110-111 Were partners all in LMIC countries, or not? Given core arguments of article, clarity around choice of which partners were contacted would be useful.

Row 136-7 Would be interesting to know how many organisations were contacted, and to discuss later the potential bias in the results/ reflect on why key informants from these eight but not other institutions responded. This is mentioned in the Limitations section, but it would add transparency to include in the Introduction or Methods the number of institutions contacted in order to see the proportion that responded positively.

Row 139 Since survey responses included individuals from institutions in the Middle East, what was the rationale behind interviewing KIs from Africa and Asia only?

Row 167 validation exercise – something a little strange about using LSTM staff to validate findings generated from interviews and survey responses from partner institutions. Implies a sense of hierarchy if LSTM had the opportunity to moderate what was said by partners, rather than the other way around, for example. Suggest adding a bit more explanation about the rationale underpinning this step.

Row 196. What about written or verbal consent for participation in interviews?

Row 211 Table 1. Use different term from ‘nationality’ as this is misleading, since the respondents nationalities are grouped into regions.

Row 229 and other quotes where ‘Senior/ ECR’ is used. Need to spell out acronym ‘ECR’ at first use, and also include seniority in Table 1 as one of the demographic indicators by which respondents were categorized. Also, why categorise KIs but not survey respondents? And what were the cut-offs for whether someone was Senior or an ECR?

Row 427. How are you differentiating between ‘fair’ and ‘equitable’? I see the terms as synonymous, and as such, using both in the same sentence is duplicative. If the authors identify a distinction, worth articulating this.

Row 451-453. Suggest a shorter description of Fig 2 – something like ‘Research-derived values of equitable partnership’ and bringing up the description of the steps towards identifying and verifying the principles into rows 451-453.

Row 463 Include reference to the TRUST Global Code of Conduct.

Row 476 Discussion section should add more references to other recent work which has elicited very similar findings and discussion points. Most of the themes presented are not original. This is not a problem since the study generates additional supporting evidence of the importance of these themes, but it seems an omission not to contextualise the study alongside other similar work.

Row 476 In the Discussion, there is not much discussion of the conflict faced by HIC institutions whereby fully embracing the principles of equitable partnership to some extent inhibits individual staff and institutional success, e.g authorship, allocation of funds etc. Worth putting this contentious challenge on the table more explicitly?

Row 525 Be explicit about which UKCDR definition is being referred to – presumably their definition of equitable partnerships? Include reference to UKCDR website/source document. Spell out acronym at this first use.

Row 649. Why a need to engage with more junior researchers but not research administrators/ managers? Were professional staff included as respondents? If so (or if not) specify in Methods section and if not, include this as a limitation since professional staff are a critical stakeholder group.

Minor points on spelling, grammar and style

Row 110. ‘Across 15 different country contexts…’. Why not just say ‘across 15 countries…’? Similar comment applies in row 136.

Row 343 A minor semantic point, but ‘virtual mediums’ sounds rather psychic. Could ‘virtual platforms’ be a better term?

Row 388 and 393 as examples. Check consistency of US v. UK spelling of words such as organization/ organisation. And Upper v lower case, e.g. row 430 v 439 northern/ Northern

Row 441 Use brackets around description of respondent. Also be consistent in whether respondent descriptor is written in italics or not.

Row 482 Commas in the wrong place. Should read ‘Research partnerships were, overall, viewed positively and equitably.

Row 594 (and elsewhere) BMJ – spell out acronym at first use.

6. PLOS authors have the option to publish the peer review history of their article (what does this mean?). If published, this will include your full peer review and any attached files.

**Do you want your identity to be public for this peer review?** For information about this choice, including consent withdrawal, please see our Privacy Policy.

Reviewer #1: No

---

## [Decision Letter · Decision Letter 1]

12 Feb 2024

PGPH-D-23-00801R1

A multi-step analysis and co-produced principles to support Equitable Partnership with Liverpool School of Tropical Medicine, 125 years on

Dear Dr. Chowdhury,

Thank you for submitting your manuscript to PLOS Global Public Health.  After careful consideration, we feel that it has merit but does not fully meet PLOS Global Public Health’s publication criteria as it currently stands. Therefore, we invite you to submit a revised version of the manuscript that addresses the points raised during the review process.

This manuscript has been passed on to me as the Academic Editor. The manuscript addresses an important topic with a critical lens, but could be strengthened greatly by addressing excellent comments from Reviewer 2. Please incorporate suggestions by Reviewer 2 and resubmit. 

We look forward to receiving your revised manuscript.

Kind regards,

Heather Haq, M.D., M.H.S.

Academic Editor

Journal Requirements:

Additional Editor Comments (if provided):

Reviewers' comments:

Reviewer's Responses to Questions

**Comments to the Author**

1. If the authors have adequately addressed your comments raised in a previous round of review and you feel that this manuscript is now acceptable for publication, you may indicate that here to bypass the “Comments to the Author” section, enter your conflict of interest statement in the “Confidential to Editor” section, and submit your "Accept" recommendation.

Reviewer #1: All comments have been addressed

Reviewer #2: (No Response)

2. Does this manuscript meet PLOS Global Public Health’s publication criteria? Is the manuscript technically sound, and do the data support the conclusions? The manuscript must describe methodologically and ethically rigorous research with conclusions that are appropriately drawn based on the data presented.

Reviewer #1: Yes

Reviewer #2: Partly

3. Has the statistical analysis been performed appropriately and rigorously?

Reviewer #1: N/A

Reviewer #2: N/A

4. Have the authors made all data underlying the findings in their manuscript fully available (please refer to the Data Availability Statement at the start of the manuscript PDF file)?

Reviewer #1: Yes

Reviewer #2: Yes

5. Is the manuscript presented in an intelligible fashion and written in standard English?

Reviewer #1: Yes

Reviewer #2: Yes

6. Review Comments to the Author

Reviewer #1: The authors have diligently and carefully responded to all my comments, and I hope they feel that the manuscript has been strengthened as a consequence. I will be very pleased to now see this in print! A great piece of work.

Reviewer #2: This is a critical topic; seeing these types of studies is excellent. Overall, it does not feel like the paper itself ‘is walking the walk’, and I would love to explore a few possible ways to help think further about the praxis of this work. Overall, there needs to be more nuanced questioning in the form of “who was not at the table” in every aspect of writing this paper.

1. Epistemic lens – The importance of epistemic diversity is mentioned (line 80), but this paper does not feel like it owns that principle in its own writing. The importance of shifting power feels like it might also be helpful to do that with the frameworks in which we write. Though the application of the RFI (Research Fairness Initiative) framework fits well here, there should be additional mentions of other frameworks derived with diverse epistemic understandings. For example, I would have loved to see a framework around collectivism or polychronic time, both often values in LMIC contexts that get disregarded with global partnerships. To strengthen this paper, it would be great to discuss some additional frameworks, why they were not used, and/or if some elements of non-Western derived frameworks could also be analyzed. Also diversify the references to include more local work.

2. Positionality statement – There are a lot of limitations of this study, which are well noted at the end. At the end of the paper, it was noted that some journals are helping with academic fairness by including ‘reflexivity statements.’ This paper could be significantly strengthened with a positionality statement for each author (or as a group) describing worldviews and proximity to power. This can help set the tone of the paper, also recognizing that this author group represents certain world views, positionality and proximity to power that makes this work only generalizable to people and populations who share such worldviews. This builds off the theme noted—specificity is critical and avoiding the over-homogenizing stereotype(s).

3. Definition of ‘Partner’- It would be really helpful to have a definition of “Partner” and “Partnership” at the beginning of the paper. It feels critical to understand how the institution and author groups define partners, as this could also add to colonial structures. As this paper illustrates, how partnerships work can create and be a product of structural violence, so using defined (and pre-existing) partnerships may be perpetuating “good feedback” bias, as these partners may also have ‘relative’ elite status and proximity to power that needs to be acknowledged.

4. Define ‘varied respondent’ – It needs to be more clear who was responding to KIIs. Please define exactly who was a varied respondent.

5. Address lack of central and S. America partnerships- This was a limitation of the study, there is a lack of geographic diversity. There are also no partners from Eastern Europe. If this is the case- critically explore why not?

6. “Data was validated” – Line 84- needs to be more detailed. Validated by what tool? Methodology here is lacking.

7. Personality differences- The discussion/theme around the importance of personality differences in PIs and its impact on partnerships is wonderful. This is addresses the critical importance of interpersonal relationships in this work. This should be more fully discussed in the discussion around accountability metrics.

8. Line 433-435 discussed the difference between academic and non-academic partnerships. This was a great point- but should also be more interrogated in the discussion with an accountability plan. Why is it that academics are more interested in outcomes? What colonial structures allow for non-academic partnerships to be more flexible?

9. Accountability- This paper needs more discussion around accountability, reparations, and what the institution owes to its ‘partners.’ These principles that were generated are great- but in order to not add more noise to the decoloniality literature, this paper needs to own some elements of accountability to change.

7. PLOS authors have the option to publish the peer review history of their article (what does this mean?). If published, this will include your full peer review and any attached files.

**Do you want your identity to be public for this peer review?** For information about this choice, including consent withdrawal, please see our Privacy Policy.

Reviewer #1: No

Reviewer #2: **Yes: **Leah Ratner

---

## [Decision Letter · Decision Letter 2]

12 Apr 2024

A multi-step analysis and co-produced principles to support Equitable Partnership with Liverpool School of Tropical Medicine, 125 years on

PGPH-D-23-00801R2

Dear Ms Chowdhury,

We are pleased to inform you that your manuscript 'A multi-step analysis and co-produced principles to support Equitable Partnership with Liverpool School of Tropical Medicine, 125 years on' has been provisionally accepted for publication in PLOS Global Public Health.

Please do make one minor grammatical change as described by one of the reviewers: "in relation to lines 88-89 where it is tautological to use the term 'partnership' in defining the term 'partner'. It would be better to use a different term in place of partnership, such as 'relationship' or a phrase such as '..that have previously worked with LSTM-Liverpool'."

Best regards,

Heather Haq, M.D., M.H.S.

Academic Editor

Reviewer Comments (if any, and for reference):

Reviewer's Responses to Questions

**Comments to the Author**

1. If the authors have adequately addressed your comments raised in a previous round of review and you feel that this manuscript is now acceptable for publication, you may indicate that here to bypass the “Comments to the Author” section, enter your conflict of interest statement in the “Confidential to Editor” section, and submit your "Accept" recommendation.

Reviewer #1: All comments have been addressed

Reviewer #2: All comments have been addressed

2. Does this manuscript meet PLOS Global Public Health’s publication criteria? Is the manuscript technically sound, and do the data support the conclusions? The manuscript must describe methodologically and ethically rigorous research with conclusions that are appropriately drawn based on the data presented.

Reviewer #1: Yes

Reviewer #2: Yes

3. Has the statistical analysis been performed appropriately and rigorously?

Reviewer #1: N/A

Reviewer #2: Yes

4. Have the authors made all data underlying the findings in their manuscript fully available (please refer to the Data Availability Statement at the start of the manuscript PDF file)?

Reviewer #1: Yes

Reviewer #2: Yes

5. Is the manuscript presented in an intelligible fashion and written in standard English?

Reviewer #1: Yes

Reviewer #2: Yes

6. Review Comments to the Author

Reviewer #1: I had previously been satisfied with the authors responses to my comments, so have no issues with this iteration - additional revisions were requested by Reviewer 2.

My only comment on this revision is in relation to lines 88-89 where it is tautological to use the term 'partnership' in defining the term 'partner'. It would be better to use a different term in place of partnership, such as 'relationship' or a phrase such as '..that have previously worked with LSTM-Liverpool'.

Reviewer #2: Tremendous, authentic and accountable re-write. I appreciate the personal reflection and break down of where global health is and where it needs to go.

7. PLOS authors have the option to publish the peer review history of their article (what does this mean?). If published, this will include your full peer review and any attached files.

**Do you want your identity to be public for this peer review?** For information about this choice, including consent withdrawal, please see our Privacy Policy.

Reviewer #1: No

Reviewer #2: **Yes: **Leah Ratner
